# Maternal Obesity and Patterns in Postnatal Diet, Physical Activity and Weight among a Highly Deprived Population in the UK: The GLOWING Pilot Trial

**DOI:** 10.3390/nu15173805

**Published:** 2023-08-30

**Authors:** Nicola Heslehurst, Emer Cullen, Angela C. Flynn, Chloe Briggs, Lewis Smart, Judith Rankin, Elaine McColl, Falko F. Sniehotta, Catherine McParlin

**Affiliations:** 1Population Health Sciences Institute, Faculty of Medical Sciences, Newcastle University, Newcastle-Upon-Tyne NE2 4AX, UK; emer.cullen@newcastle.ac.uk (E.C.); judith.rankin@newcastle.ac.uk (J.R.); elaine.mccoll@newcastle.ac.uk (E.M.); falko.sniehotta@newcastle.ac.uk (F.F.S.); catherine.mcparlin@newcastle.ac.uk (C.M.); 2Department of Nutritional Sciences, Faculty of Life Sciences and Medicine, King’s College London, London WC2R 2LS, UK; angela.flynn@kcl.ac.uk

**Keywords:** diet, physical activity, weight, obesity, pregnancy, postnatal, deprivation, UK

## Abstract

Preconception obesity is associated with adverse pregnancy outcomes and deprivation. The postnatal period provides an opportunity for preconception intervention. There is a lack of published postnatal behaviour and weight data to inform intervention needs. Secondary analysis of the GLOWING study explored postnatal diet, physical activity (PA) and weight among women living with obesity in deprivation. Thirty-nine women completed food frequency and PA questionnaires and provided weight measurement(s) between 3–12 months postnatal. Women’s diet and PA fell short of national guidelines, especially for fruit/vegetables (median 1.6–2.0 portions/day) and oily fish (0–4 g/day). PA was predominantly light intensity. Patterns in weight change across time points indicated postnatal weight loss compared with 1st (median −0.8 to −2.3 kg) and 3rd-trimester weights (−9.0 to −11.6 kg). Weight loss was higher among women without excessive gestational weight gain (GWG) (−2.7 to −9.7 kg) than those with excessive GWG (2.3 to −1.8 kg), resulting in postnatal weight measurements lower than their 1st trimester. These pilot data suggest preconception interventions should commence in pregnancy with a focus on GWG, and postnatal women need early support to achieve guideline-recommendations for diet and PA. Further research in a larger population could inform preconception intervention strategies to tackle inequalities in maternal obesity and subsequent pregnancy outcomes.

## 1. Introduction

The association between pre-pregnancy obesity and adverse maternal and infant health outcomes in pregnancy and throughout the life course is well-established, highlighting the need for preconception weight management support. Complications include hypertensive disorders (e.g., preeclampsia), gestational diabetes, caesarean deliveries, stillbirth, maternal mortality, congenital anomalies, and macrosomia [1,2,3]. Maternal obesity is also a leading risk factor for the development of child obesity [4]. One in five women in the United Kingdom (UK) enter pregnancy with a body mass index (BMI) in the obese range (BMI ≥ 30 kg/m^2^) [5], and deprivation is a key driver of inequalities in prevalence [6]. During pregnancy, gestational weight gain (GWG) is a natural response, accommodating the growing fetus [7]. However, over half of women exceed GWG recommendations, with the highest prevalence amongst women with an overweight and/or obese BMI before pregnancy [8]. Often, GWG is not lost following pregnancy, and some women continue to gain further weight during the postnatal period, increasing the risk of overweight and obesity in subsequent pregnancies [9,10,11].

Supporting women with weight management in the postnatal period is an opportunity for preconception intervention for subsequent pregnancies—the inter-pregnancy period. A recent systematic review showed that, for women with overweight or obesity, interpregnancy weight loss decreased the risk of caesarean delivery, gestational diabetes and hypertensive disorders in subsequent pregnancies [12]. Current recommendations advise that women living with overweight or obesity safely lose weight during the postnatal period and before subsequent pregnancies [13,14,15]. In the UK, postnatal women, up to 12 months postpartum, are recommended to achieve 150 min of moderate intensity physical activity (PA) each week and twice weekly strengthening activities [16], with a focus on listening to their body, taking a gradual approach to increase activity, and incorporating pelvic floor exercises. Dietary recommendations [17] advise the consumption of a varied and healthy diet, as outlined in the Eatwell Guide [18], including five portions of fruits and vegetables daily, fibre-rich foods, low-fat alternatives, basing meals on starchy/wholegrain foods, two portions of fish weekly (one oily), substitution of red/processed meat for pulses, beans and lentils, and avoidance of foods high in fat, salt and sugar. For women exclusively breastfeeding, the National Institute for Health and Care Excellence (NICE) guidelines recommend an additional 330 kcal/day for the first 6-months and 400 kcal/day for the second 6-months [15].

Despite the existence of guidelines, there are no published data reporting patterns of postnatal PA, diet, and weight change in women with obesity in the UK or globally. A 2015 systematic review, not restricted to women with obesity [19], found that PA levels decreased over the course of pregnancy and did not typically return to pre-pregnancy levels in the first year postnatal. An Australian study [20] among women with overweight or obesity also reported PA levels to decline over the course of pregnancy, with an increase during the postnatal period; however, by four months postnatal, PA levels had not reached the levels reported in early pregnancy [21]. There is a lack of systematic reviews reporting postnatal diet patterns. Studies from the USA [22,23] and Australia [20] report low-quality postnatal diets among women with overweight or obesity, with low consumption of fruit, vegetables, wholegrains and milk, and diets high in sodium, saturated fats and calories. There is a lack of published data on postnatal dietary patterns among women in the UK. Similarly, there is a lack of data published in the UK relating to postnatal patterns in weight change. One study published in 2000 [24] explored patterns in weight change from 13 weeks’ gestation to 6-months postnatal in 17 women with obesity. They reported a mean change of 0.6 kg (SD 6.4), with most women “considerably heavier” at 6-months postnatal than 13-weeks’ gestation.

Gaining a better understanding of women’s dietary and PA behaviours’ and weight change during the postnatal period would highlight areas where women might benefit from additional intervention and support to improve their overall health and contribute to healthy weight management. This study aimed to explore the patterns of diet PA behaviours, and weight change among postnatal women living with obesity in a highly deprived region of the UK.

## 2. Materials and Methods

A secondary analysis of data collected as part of the GLOWING pilot trial was undertaken. GLOWING was developed to support midwives’ implementation of UK guidelines for weight management during pregnancy; full details of the intervention are published elsewhere [25]. This cluster randomised controlled trial involved four NHS trusts (i.e., clusters) in northeast England. The intervention and control arms each had two large and one small clusters, based on annual birth rates for each NHS Trust. The intervention content and delivery methods were designed using social cognitive theory. In the intervention arm, small groups of midwives received an intensive full-day face-to-face training session and provided information resources to share with women in their routine practice. The control arm received no intervention.

The northeast of England has a high prevalence of both deprivation and maternal obesity [26,27]. In 2018/2019, 27.4% of pregnant women in the northeast had obesity compared with the England average of 22.1% [28]. Whilst the primary outcome of GLOWING was midwives’ behaviour, secondary outcomes aimed to explore patterns of diet and PA behaviours, and weight changes among pregnant women living with obesity. Pregnant women recruited to GLOWING did not directly receive any intervention. Women were eligible if they had a 1st trimester BMI ≥ 30.0 kg/m², were over 18 years of age and could speak and read English. Following the intervention delivery, 104 women were recruited at 12-week’s gestation to provide socio-demographic data and data regarding discussions they had with their midwives at their first antenatal contact. This sample size was determined using recommendations for pilot trials [25]. Women received a £10 gift voucher for each questionnaire returned. The women were requested to complete a questionnaire reporting their diet, PA, and weight at 36-weeks’ gestation and 3-, 6-, 9- and 12-months postnatal. The pregnancy dietary, PA and weight data have been published [29]. This paper reports the postnatal data. The pilot trial was not powered to detect differences between intervention and control groups [25], and analysis confirmed no differences in postnatal outcomes between intervention arms (Appendix A). Therefore, the remaining analyses pooled intervention and control data together. The STROBE guidelines for reporting observational studies have been used (Appendix A). The pilot study received ethical approval on 16 December 2015 [25].

### 2.1. Socio-Demographic Data Collection

The socio-demographic questions include both quantitative and free-text items relating to the women’s 1st trimester BMI, ethnic group, employment, marital, smoking, and breastfeeding status, alcohol intake, and dietary preference. Women’s 1st trimester BMI were collected from electronic health records and used to categorise their obesity class using the WHO criteria: class 1 BMI 30.0–34.9 kg/m^2^, class 2 BMI 35.0–39.9 kg/m^2^, and class 3 BMI ≥ 40 kg/m^2^ [30]. A woman’s postcode of residence was linked to the Index of Multiple Deprivation (IMD) quintiles and ranked, with quintile 1 being the most deprived and quintile 5 the least deprived [31].

### 2.2. Diet, Physical Activity and Weight Data Collection

Both the postnatal diet and PA data collection and coding replicated the methods used during the pregnancy phase of the study, which have been published [29]. In brief, for diet, a 50-item semi-quantitative food frequency questionnaire (FFQ) [32,33] was used, including free-text responses (brand and monthly consumption of bread, cereal, butter/spread and cheese) and a multiple-response grid to report the intake of various food items over the month. Dietary data were coded to estimate portion sizes and amounts of food items consumed each day.

PA data were collected using the 32-item validated Pregnancy PA Questionnaire (PPAQ), including questions about the type and duration of activity in the previous week [34]. These were grouped into PA domains (Household/Caregiving, Inactivity; Occupational, Sports/Exercise, and Transportation) and metabolic equivalents (MET) values were estimated to classify activities into different intensities: Sedentary (<1.5 METs); Light (1.5–3.0 METs); Moderate (3.0–6.0 METs); and Vigorous (>6.0 METs) [35].

Women were asked to have their weight measured by a health professional at each postnatal questionnaire time point (3-, 6-, 9- and 12-months) to be provided to the research team or to provide a printed copy of their weight measurement from scales in health centres or pharmacies at each time point. We could not access postnatal weight measurements from medical records as these are not routinely collected. We used pregnancy weight measurements to estimate weight change at each postnatal time point from the 1st-trimester booking appointment (mean gestational age 11.2 weeks, SD 4.1) and the 3rd trimester (36 weeks’ gestation). Pregnancy weight data collection methods have been published [29] and included a combination of weights measured for the research and from routine medical records. GWG was defined as adequate (i.e., within guideline recommendations [7]), inadequate (i.e., below guideline recommendations), or excessive (i.e., above guideline recommendations). Due to the limited number of women whose GWG was defined as adequate, the inadequate and adequate categories were merged to create a non-excessive GWG category [29].

### 2.3. Data Analysis

Not all participants completed each follow-up questionnaire. Therefore, cross-sectional analysis was undertaken at each postnatal time point. As these are pilot data, descriptive analysis was carried out on women’s diet, PA, and weight change data at 3-, 6-, 9- and 12-months postnatal to explore patterns. Comparison of behaviours between categories of obesity class, breastfeeding status, deprivation status, ethnicity, and dietary preference (i.e., vegetarians vs. non-vegetarians) was planned to be undertaken, but due to the small sample size and limited numbers within some sub-categories, this was not possible. The outcome data were tested for normality using the Shapiro-Wilkes test; data that were normally distributed are presented as means and standard deviations (SD), and data that are non-normally distributed as median and interquartile ranges (IQR). Diet data reported are for food types, with average daily consumption (e.g., mL/day milk consumed) and number and percent consuming the food item type (e.g., reduced fat, full fat). PA outcomes are average total energy expenditure (EE) per week and EE in each intensity category and from each PA domain. Weight change data reported are for average change at each postnatal time point from both 1st and 3rd trimesters of pregnancy, as well as a comparison for average weight change among women with excessive or non-excessive GWG.

## 3. Results

### 3.1. Participants Characteristics

In total, 39 women who returned pregnancy questionnaires also returned postnatal questionnaires at one or more time points (38%) between March 2018 and June 2019. At 3-, 6-, 9- and 12-months postnatal, the socio-demographic, diet and PA data were available for 24 (23.1%), 22 (21.2%), 12 (11.5%) and 20 (19.2%) women, respectively. Weight data were available for 20 (19.2%), 22 (21.2%), 11 (10.6%) and 20 (19.2%) women respectively. At nine months postnatal, one NHS trust did not send out any questionnaires, which explains the relatively low response at this postnatal time point compared with others. There were no significant differences in the socio-demographic characteristics of the women who returned the questionnaire compared to those who did not (Appendix A).

Across the four postnatal time points, there was minimal difference in the median BMI, ranging from 35.0 to 36.6 kg/m^2^ and most women had class 1 or 2 obesity (Appendix A). The mean age was similar across all time points, ranging from 29.0 to 31.2 years (reflective of the national average) [36]. Participants had a median of two pregnancies in total, and lasting >24 weeks, at all-time points. Most women resided in the most deprived quintiles, with 45.8–65.0% in Quintile 1 and 15.0-25.0% in Quintile 2, higher than the national average (23.2% and 22.5% respectively) [36]. Most women were white (83.3–95.0%, higher than the national average of 81.6% [36]) and were in some form of paid employment (60.0–81.8%, reflective of the national average [37]). Education varied across the time points; however, most women attained at least GCSEs or equivalent (i.e., high school level, 79.5–100.0%). There were 50.0–63.6% of women married and 31.8–45.8% single (lower than the national average of 57% single [38]). Smoking status was similar across all time points, with 75.0–86.3% of women reporting not smoking. This demonstrates a higher proportion of smokers in the GLOWING study than the UK adult female population (rate of 11.5% [39]). At 3-, 6-, 9- and 12 months postnatal, 20.8%, 27.3%, 8.3% and 0% of women were exclusively breastfeeding or breastfeeding with formula feeding. Regarding dietary preferences, most women consumed meat and fish, ranging from 70.8–85.0%, and no women followed a vegetarian diet. The number of women consuming alcohol varied across time points (35.0–75.0%), with a median intake of 2–3 units/week.

### 3.2. Dietary Behaviours

Across the postnatal time points, the median milk intake was 142 mL/day, spread ranged from 5–10 g/day, and cheese 4–12 g/day (Table 1). There was a pattern towards the GLOWING participants consuming reduced fat options for milk and spread but not for cheese. The median intake of sugary drinks was highest at 3-months postnatal (158 mL/day) compared to the other time points, particularly for the intake of sugar-sweetened beverages. The median intake of starchy carbohydrates was low, ranging from 85–109 g/day. Most women consumed 1–2 slices of bread daily, with approximately even distribution between wholemeal and white varieties. Around half (41.7–65.0%) of women did not consume breakfast cereals, but those who did were more likely to consume non-refined types. The median intake of snacks was similar across the time points, ranging from 31–61 g/day, with the median intake of sweet snacks being the highest. Compared to national recommendations, women’s intake of fruit and vegetables was low; daily intake ranged from 1.6–2.0 portions/day. Women’s consumption of meat and fish tended to be primarily from red meat and processed meat and fish sources, and intake of unprocessed fish was low (16–37 g/day), specifically oily fish, with median intakes ranging from 0–4 g/day.

### 3.3. Physical Activity Behaviours

Reported levels of PA showed a pattern towards higher median EE with each postnatal time point, from 213.4 MET-hr/week at 3-months postnatal to 300.7 MET-h/week at 12-months (Table 2). In relation to PA intensity, most EE was from light-intensity PA across all time points, followed by moderate intensity, with very few MET-hours/week from vigorous PA. EE from sedentary PA remained similar at 17.9 MET-h/week across all time points. In relation to PA domains, the majority of EE was from participating in household/care activities across all time points (131.9–180.1 MET-h/week), with lowest EE from the occupation (0 MET-hr/week) and sport (0.5–5.3 MET-h/week) domains.

### 3.4. Postnatal Weight Change

All women in this study had their weight recorded at their booking appointment, and at least one postnatal weight was available for 35 women. The last recorded postnatal weight for 21 women (60%) was lower than their booking weight. When looking at weight change, the median values across all time points showed weight loss compared to both booking (−0.8 to −2.3 kg) and 3rd-trimester weight (−9.0 to −11.6 kg), although the IQRs showed that some women gained weight (Table 3). Median weight was lowest at 3-months postnatal (95.1 kg), gradually increasing across each postnatal period to the highest median postnatal weight at 12-months (98.0 kg, Figure 1). When comparing postnatal weight change between women whose GWG was excessive or not excessive, median weight change (loss) from booking was greater across all time points for those without excessive GWG (−2.7 to −9.7 kg) than those with excessive GWG (2.3 to −1.8 kg) (Table 3). When comparing weight change from 3rd trimester, median weight loss at each time point is similar between those with excessive GWG (−9.0 to −11.9kg) and not excessive GWG (−6.1 to −11.5 kg). These data suggest that having a GWG that is not excessive may support women with obesity in achieving a greater postnatal weight loss that results in lower weight than their booked weight.

## 4. Discussion

This secondary analysis has provided novel data on postnatal diet, PA, and weight change among a highly deprived population of women living with obesity in the UK. This population had a suboptimal dietary intake, particularly regarding fruit and vegetables, unrefined breakfast cereals, fish, and oily fish, which did not meet national recommendations. Additionally, guideline recommendations are to limit the consumption of red meat and processed meat and fish, with no more than 70 g/day, whereas the median intake in this population was higher than these recommendations at most postnatal time points. Light-intensity PA contributed most to overall EE across all time points, whereas guideline recommendations are for at least 150 min/week of moderate-intensity PA. Overall, most EE was from the household/care PA domain, suggesting this might be an area for interventions to focus on, for example, supporting women to increase the intensity of activity in domains they are already participating in. There was a pattern towards weight loss across all postnatal time points when compared with 1st- and 3rd-trimester weights, with the greatest weight loss observed at 3-months postnatal and among women whose GWG was not excessive. Interventions could focus on limiting GWG as part of a preconception strategy for subsequent pregnancies and supporting women to maintain early postnatal weight loss from three months and throughout the 12-months.

There is little existing evidence exploring postnatal diets of women in the UK. However, the findings in this study are similar to international studies in both the USA [22,23] and Australia [20]. These studies also found suboptimal postnatal diets in women with obesity, with low intakes of fruit and vegetables [20,22,23], and diets low in wholegrains [20,22]. High intakes of red meat and processed meat and fish, along with food generally high in saturated fats and salt, were also reported [22]. However, this population’s reported low intake of fish and oily fish was not reflected in other studies but may reflect the low intake of oily fish reported in the general UK population in the UK National Diet and Nutrition Survey [40]. The dietary findings in this study are similar to those found in the published study exploring pregnancy behaviours [29]. Although longitudinal analysis was not undertaken, the findings may be suggestive that women’s diet behaviours do not substantially change when transitioning between pregnancy and postnatal periods. This is supported by an Australian study [20], which reports that the low diet quality of women with obesity during pregnancy is sustained throughout the postnatal period.

Furthermore, Stephenson et al. [41] summarised intervention evidence showing that positive dietary changes in pregnancy may result in sustained positive dietary behaviours throughout the postnatal period. This further highlights the opportunities for preconception interventions to commence in pregnancy and be continued in the postnatal period to benefit subsequent pregnancies. Diet patterns during the postnatal period are also associated with future eating behaviours of children [42]; therefore, interventions to optimise postnatal diet may have life course benefits for both women and their children, as well as for subsequent pregnancies.

There is a lack of research in the UK focusing on women living with obesity in the postnatal period to compare our findings with. However, our findings are similar to those reported in the general postnatal populations. Our study showed increasing median EE over the postnatal period, and an Australian cross-sectional study also showed that sitting time was highest and PA levels lowest in the first six months postnatal, increasing significantly between 6–12 months [43]. In our study, the overall EE reported at each time point in the postnatal period was higher than that reported in the GLOWING cohort in pregnancy (166 MET-h/week, IQR 128.1, 249.2) [29], suggesting an increase in PA following pregnancy. Postnatal PA intensity and domains followed a similar pattern to pregnancy, suggesting that while overall EE might be lower in pregnancy than postnatal, the proportional intensities and types of activities are similar. Occupational activity was the exception, explained by some women in the GLOWING population working during pregnancy and being on maternity leave postnatally.

In our study, there was an overall pattern for weight loss in the postnatal period compared with booking or 3rd-trimester weights, and 60% of women had postnatal weights lower than their booking weight. These patterns are similar to a secondary analysis of the UK UPBEAT trial in women with obesity, which showed 52% had a lower weight at 6-months postnatal than in their 1st-trimester [44]. However, these results differ from an observational study in the UK among women with obesity, which showed a mean weight gain at 6-months postnatal and most women with higher postnatal weight than booking weight [24]. Our study and the UPBEAT trial were conducted around 20 years after this previously published study, and there has been the publication of maternal obesity and weight management guidelines in this time period [15], making addressing maternal obesity and weight management more prominent in routine maternity care. This may have contributed to greater awareness of weight management among health professionals and women and resulted in patterns towards increased postnatal weight loss in the UK. Additionally, the small sample sizes for postnatal weight measurements in our study and the previously published observational study (*n* = 17) may have contributed to the differences observed, and larger longitudinal observational studies are needed to explore patterns in postnatal weight over time. However, it is important to note that we did observe a clear difference in the amount of weight loss when comparing GWG groups. These findings reflect other studies showing that excessive GWG is associated with weight retention in the postpartum period [44,45]. The UPBEAT trial included pregnant women living with obesity and found that GWG < 9 kg, higher levels of PA and exclusive breastfeeding for more than four months were associated with negative postpartum weight retention [44].

In our study, we could not explore postnatal behaviour and weight change patterns according to breastfeeding status as planned due to too few women breastfeeding. At 3-months postnatal, only 20% were breastfeeding, and 27% at 6-months, lower than the population average outlined in the latest UK-wide infant feeding survey (34% at 6-months) [46]. Generally, women with obesity are less likely to breastfeed, and those who do are at an increased risk of early breastfeeding cessation [2,47]. In addition to obesity, there are numerous socio-demographic barriers to breastfeeding that may be present in this population. The northeast of England has very high levels of deprivation and consistently lower rates of breastfeeding compared to the national average, with the most recent data showing prevalence at 6–8 weeks of 35.7% (95% CI 35.2–36.4) compared to 49.5% (95% CI 49.1–49.3) nationally. Women are more likely to breastfeed from areas of low deprivation, older and from minority ethnic groups [48], whereas the GLOWING population were highly deprived and predominantly white, living in a region with low levels of breastfeeding, which may impact social and cultural norms. It is well-established that breastfeeding has a plethora of benefits. Breastfeeding is recommended as part of postnatal weight management strategies (along with diet and PA behaviours) and is associated with reduced prevalence of childhood obesity [15,49]. This population of women would highly benefit from breastfeeding support for their health, as well as the health of their infant. However, this requires further investigation to understand and overcome any barriers to breastfeeding that may be present in this population of women.

### Strengths and Limitations

This is one of the first studies to explore the postnatal dietary and PA behaviours of women living with obesity in an area of high deprivation in the UK. Extensive data were collected as part of the GLOWING pilot trial, providing a rich dataset for analysis. However, this study was a secondary data analysis with a small sample size. Therefore, it was not designed to be an observational study or powered to detect statistical differences or make inferences at the population level. Loss to follow-up was high, and longitudinal analysis was not possible. However, there were limited statistically or clinically significant differences in the socio-demographics of women who returned questionnaires compared with those lost to follow-up. The sample sizes limited the ability to explore diet and PA behaviours or weight change between population subgroups, including by ethnic group, obesity category, or deprivation. Within the study population, women were predominantly white, employed, and could speak English, which may not reflect other UK regions. 

Data collection using FFQs and PPAQ has benefits in relation to low-resource options to reach many participants. However, both are subjective, relying on memory, literacy and numeracy skills and are subject to recall bias [50], and some self-reported diet values appear to be lower than expected. Consequently, these surveys may not provide the most accurate representation of individual diet or PA behaviours. Other data collection options (e.g., food diaries or activity monitors such as accelerometers) are alternative options for these measurements. However, this pilot study’s choice of data collection methods needed to consider participant burden and the feasibility of data collection tools in a larger definitive trial. As the wider pilot study data collection included extensive questionnaires, the robustly validated FFQ [44] and PPAQ [34] were considered the most appropriate methods to be embedded within trial procedures and to limit participant burden. The potential under-reporting in this study also reflects under-reporting in other studies, with the highest prevalence of under-reporting among women with pre-pregnancy overweight and obesity [51,52].

This study identified many areas for future research, along with reflections from the research team on resolving some of the issues relating to loss to follow-up. Pregnancy and the postnatal periods are times of transition, including moving home due to a growing family. We lost some participants to follow up due to this (e.g., questionnaires were returned with “no longer at this address” written on them), and some women whom we had alternative contact details (e.g., those consenting to be interviewed as well as completing questionnaires provided email addresses or phone numbers) reported that they had moved and had not received the questionnaire. In the questionnaires, women were asked to state if they preferred paper or online versions, and paper was the main preference. However, this method does not have the flexibility that electronic follow-up would have, as email addresses are less likely to change. 

Additionally, posting questionnaires for follow-up was led by busy research midwife teams within the NHS Trusts, including one Trust that did not send out any questionnaires during the 9-month follow-up period. In future studies, keeping consent for contact details with university research teams for follow-up may be beneficial. Reducing loss-to-follow-up would allow a longitudinal study to be undertaken, offering important insights that could inform the development of personalised preconception interventions.

## 5. Conclusions

The postnatal period is important for the mother’s long-term health, as well as preconception health for those who have subsequent pregnancies. This descriptive study suggests that, among a highly deprived population of women living with obesity in the UK, postnatal diet and PA behaviours are inadequate and do not meet guideline recommendations. However, while there is variation at the individual level, the patterns in postnatal weight loss are promising, and suggest that strategies to limit excessive GWG and to support early postnatal weight loss may be beneficial for women to achieve postnatal weights lower than their 1st-trimester weight; therefore, having an impact on subsequent pregnancies. These data support the need to focus on the postnatal period as a preconception period. However, early provision of preconception support should also be extended to pregnancy with a more joined-up approach between pregnancy and postnatal stages. Further longitudinal research is required to explore these findings in a larger population of deprived women living with obesity to better understand postnatal behaviours and weight patterns and to inform the development of interventions to tackle inequalities in maternal obesity and subsequent pregnancy outcomes.

## Figures and Tables

**Figure 1 nutrients-15-03805-f001:**
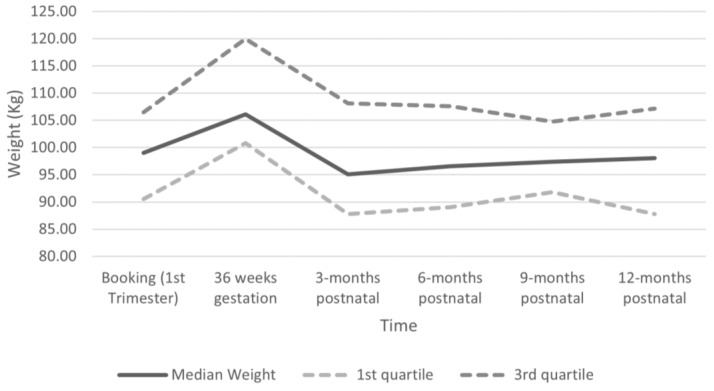
Median Weight from Booking to 12 Months Postnatal. Booking (1st trimester): *n* = 35, median 99.0 kg (IQR 90.6, 106.5); 36 weeks gestation: *n* = 25, median 106.1 kg (IQR 100.8, 120); 3-months postnatal: *n* = 20, median 95.1 kg (IQR 87.8, 108.2); 6-months postnatal: *n* = 22, median 96.5 kg (IQR 89.1, 107.6); 9-months postnatal: *n* = 10, median 97.4 kg (IQR 91.8, 104.8); 12-months postnatal: *n* = 20, median 98.0 kg (IQR 87.8, 107.2).

**Table 1 nutrients-15-03805-t001:** Maternal diet at 3-, 6-, 9- and 12-months postnatal (Median, IQR unless specified).

	3-Months	6-Months	9-Months	12-Months
Milk mL/d	142 (142, 285)	142 (142, 285)	142 (142, 285)	142 (36, 285)
*Reduced fat (n, %)*	19 (79.2)	18 (81.8)	8 (66.7)	11 (55)
*Full fat (n, %)*	3 (12.5)	3 (13.6)	3 (25)	5 (25)
*None (n, %)*	2 (8.3)	1 (4.5)	1 (8.3)	4 (20)
Spread g/d	6 (0, 11)	9 (4, 23)	10 (5, 14)	5 (1, 9)
*Reduced fat (n, %)*	11 (45.8)	13 (59.1)	6 (50)	12 (60)
*Full fat (n, %)*	7 (29.2)	8 (36.4)	5 (41.7)	4 (20)
*None (n, %)*	6 (25)	1 (4.5)	1 (8.3)	4 (20)
Cheese g/d	4 (0, 30)	9 (0, 36)	4 (0, 22)	12 (0, 25)
*Reduced fat (n, %)*	3 (12.5)	3 (13.6)	2 (16.7)	2 (10.0)
*Full fat (n, %)*	12 (50.0)	13 (59.1)	6 (50.0)	12 (60.0)
*None (n, %)*	9 (37.5)	6 (27.3)	4 (33.3)	6 (30.0)
Sugary Drinks mL/d	158 (40, 500)	80 (3, 279)	14 (0, 211)	57 (14, 518)
*Fruit juice*	0 (0, 27)	0 (0, 28)	7 (0, 24)	14 (0, 24)
*Sugar-sweetened beverages*	151 (0, 490)	76 (0, 226)	0 (0, 178)	10 (0, 500)
Starchy carbohydrate foods g/d	103 (62, 140)	93 (71, 158)	109 (88, 160)	85 (29, 149)
*Rice, pasta, noodles, potatoes*	14 (12, 17)	16 (13, 18)	16 (11, 20)	15 (13, 19)
*Takeaway and oven chips*	4 (3, 6)	5 (4, 6)	5 (3, 6)	4 (4, 6)
Bread g/d	51 (26, 76)	54 (31, 90)	72 (57, 126)	36 (12, 72)
*Wholemeal bread (n, %)*	12 (50.0)	11 (50.0)	5 (41.7)	10 (50.0)
*White bread (n, %)*	9 (37.5)	10 (45.5)	7 (58.3)	7 (35.0)
*No bread (n, %)*	3 (12.5)	1 (4.5)	0	3 (15.0)
Breakfast cereal g/d	10 (0, 30)	0 (0, 38)	0 (0, 20)	0 (0, 29)
*Refined breakfast cereal (n, %)*	7 (29.2)	3 (13.6)	3 (25)	2 (10.0)
*Non-refined breakfast cereal (n, %)*	7 (29.2)	6 (27.3)	1 (8.3)	5 (25.0)
*No breakfast cereal (n, %)*	10 (41.7)	13 (59.1)	8 (66.7)	13 (65.0)
Fruits and Vegetables g/d	161 (80, 355)	147 (58, 256)	131 (41, 306)	136 (47, 286)
*Vegetables*	89 (33, 178)	78 (42, 132)	75 (41, 306)	68 (22, 115)
*Fruits*	50 (20, 200)	81 (17, 152)	67 (35, 156)	45 (16, 137)
*Servings of fruit and vegetables per day* ^a^	2.0	1.8	1.6	1.7
Snacks g/d	43 (26, 84)	51 (24, 93)	61 (27, 145)	31 (14, 70)
*Crisps and fried snacks*	3 (2, 4)	5 (4, 5)	4 (2, 5)	2 (2, 5)
*Sweet snacks*	33 (22, 75)	45 (18, 84)	52 (18, 140)	26 (9, 64)
*Yoghurt*	3 (1, 5)	3 (1, 4)	3 (1, 6)	2 (1, 4)
Meat and fish g/d	151 (93, 222)	169 (128, 195)	147 (95, 197)	124 (85, 169)
*Red meat*	21 (11, 68)	68 (22, 96)	68 (3, 68)	22 (14, 68)
*Processed meat and fish*	19 (19, 54)	28 (0, 55)	19 (19, 38)	19 (10, 35)
*Fish (including processed)*	16 (7, 32)	17 (0, 35)	37 (15, 53)	23 (0, 35)
*Oily fish*	0 (0, 7)	0 (0, 13)	4 (0, 17)	0 (0, 9)

^a^ Servings of fruit and vegetables calculated based on 80 g serving size, e.g., median intake of 161/80 = 2.0 portions per day.

**Table 2 nutrients-15-03805-t002:** Maternal physical activity at 3-, 6-, 9- and 12-months postnatal.

	MET-h/Week Median (IQR)			
	3-Months	6-Months	9-Months	12-Months
Total (EE)	213.4 (137.9–295.5)	224.3 (163.8–292.1)	239.3 (163.2–313.2)	300.7 (198.2–415.5)
Sedentary PA	17.9 (14.9–29.4)	17.9 (7.4–18.1)	17.9 (17.9–29.4)	17.9 (7.4–29.3)
Light PA	102.5 (76.4–150.9)	111.3 (75.3–155.4)	111.7 (93.4–146.0)	146.0 (105.4–204.0)
Moderate PA	89.6 (51.1–115.1)	89.2 (53.8–138.4)	76.6 (40.2–127.3)	109.6 (66.5–210.4)
Vigorous PA	0.8 (0.0–0.8)	0.0 (0.0–0.8)	0.0 (0.0–0.0)	0.0 (0.0–1.4)
Household/ care PA	166.6 (102.84–212.2)	180.1 (107.4–234.2)	131.9 (101.2–180.1)	163.5 (107.4–238.0)
Occupational PA	0.0 (0.0–0.0)	0.0 (0.0–0.0)	0.0 (0.0–88.9)	0.0 (0.0–123.5)
Sport PA	5.3 (1.7–13.6)	4.0 (1.3–5.5)	0.5 (0.0–5.3)	1.6 (0.1–6.4)
Transport PA	28.0 (10.2–43.3)	23.3 (10.5–31.5)	13.4 (10.7–22.6)	17.4 (10.7–40.7)
Inactive PA	21.5 (16.7–32.8)	17.9 (7.4–22.0)	29.4 (17.9–30.5)	17.9 (7.4–34.5)

Abbreviations: METs = Metabolic Energy Equivalents, EE = energy expenditure, and PA = physical activity.

**Table 3 nutrients-15-03805-t003:** Postnatal weight change at 3-, 6-, 9- and 12-months, from booking and 3rd-trimester weights.

		Median (IQR) Postnatal Weight Change			
Population	Weight Change Period	3 Months	6 Months	9 Months	12 Months
All women	From booking	−2.3 (−9.7, 1.6)	−1.3 (−6.9, 4.8)	−0.8 (−3.5, 3.2)	−1.6 (−8.8, 3.3)
	From 3rd trimester	−11.6 (−15.6, −5.2)	−10.2 (−12.7, −5.0)	−9.0 (−11.2, −6.0)	−9.0 (−10.8, −4.1)
GWG “not excessive”	From booking	−9.7 (−13.5, −2.2)	−8.0 (−10.8, −2.5)	−2.7	−4.4 (−9.4, 3.6)
	From 3rd trimester	−11.5 (−13.7, −4.2)	−10.2 (−12.7, −5.4)	−6.1	−7.2 (−12.0, −0.6)
GWG “excessive”	From booking	−1.8 (−2.5, 4.2)	2.3 (−3.4 6.0)	−0.8 (−3.0, 0.8)	−0.3 (−7.1, 2.2)
	From 3rd trimester	−11.9 (−16.1, −9.4)	−10.1 (−13.3, −4.2)	−9.8 (−11.1, −6.8)	−9.0 (−18.9, −5.8)

Weight measured at booking is in the 1st trimester of pregnancy. Weight measured in the 3rd trimester of pregnancy was approximately 36 weeks gestation. GWG “not excessive” refers to weight gain within or below guideline recommendations [7]. Abbreviations: GWG = gestational weight gain; IQR = interquartile range.

## Data Availability

The data presented in this study are available on request from the corresponding author. The data are not publicly available due to ethical approval restrictions, and any further data sharing will be subject to necessary approvals.

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
