# Peer review of "Maternal Obesity and Patterns in Postnatal Diet, Physical Activity and Weight among a Highly Deprived Population in the UK: The GLOWING Pilot Trial"

_nutrients, 2023, doi:10.3390/nu15173805_

Round 1

Reviewer 1 Report

The purpose of this paper was to explore the pattens of diet and physical activity in women with obesity living in the deprivation in the UK. It was stated that this was a secondary analysis of data obtained from a midwife study. The authors presented the results in easy to understand tables and graphs. The one limitation, per the authors, was the small sample size but this is understandable as it is incredibly difficult to keep participants in long-term studies. I do hope the authors will continue to follow up with a breastfeeding(BF) related study as BF has a great impact on weight loss post-pregnancy.

Overall, good manuscript and of interest to those who provide care and guidance to mothers.

Author Response

The purpose of this paper was to explore the patterns of diet and physical activity in women with obesity living in the deprivation in the UK. It was stated that this was a secondary analysis of data obtained from a midwife study. The authors presented the results in easy to understand tables and graphs.

  • Thank you for this positive feedback

The one limitation, per the authors, was the small sample size but this is understandable as it is incredibly difficult to keep participants in long-term studies.

  • We agree that there are substantial challenges with postnatal follow up and have discussed these, with recommendations on how to improve on loss to follow up in future studies.

I do hope the authors will continue to follow up with a breastfeeding(BF) related study as BF has a great impact on weight loss post-pregnancy.

  • Thank you for this feedback. There is much research still to do relating to weight management in the preconception, pregnancy and postnatal period, and breastfeeding is an important part of this research. Our study identified the very low rates of breastfeeding in this highly deprived population of women living with obesity, which warrants further exploration to improve breastfeeding rates for both maternal and infant life course health.

Overall, good manuscript and of interest to those who provide care and guidance to mothers.

  • Thank you for this positive feedback, we agree that this manuscript is highly relevant to practitioners and researchers in this field, providing novel data for an under-explored and highly vulnerable population.

Reviewer 2 Report

Heslehurst et al present a secondary analysis of a study of postnatal diet, physical activity (PA) and body weight in women with obesity from deprived areas in the UK. This manuscript extends that study (which was published in Nutrients) with follow-up data from the 3, 6, 9 and 12 months postnatal timepoints. These new data aimed to deepen the understanding of women’s diet, PA and weight changes in the postnatal period as it can be considered to also be a prenatal period for subsequent births. That time period is thus a window of opportunity for establishing improved health and lifestyle during the conceptional and gestation stages for subsequent children. Additionally, this type of study has not been performed before in the UK so the data will be valuable for that population.

Major comments:

1)      As the authors repeatedly acknowledge, the study suffers from low subject numbers at every stage. While the study was undertaken with adherence to standard design guidelines, the low participant numbers make it difficult to justifiably extrapolate to larger populations. Statistical significance for novel observations is largely absent, even for the intriguing effect of gestational weight gain on postnatal weight loss (Table 3). Overall, this study supports the general view that future research should concentrate on obese women from deprived areas, but these results don’t add much to the current body of evidence.

2)      The conclusions state that the study has 'identified' that postnatal diet and PA are inadequate - I don't see that the evidence does more than 'suggest' this - even though we know it to be true from other research.

3)      The reported dietary intake data are surprisingly low. Measuring dietary intake is notoriously difficult, one of the most important issues being the accuracy of dietary recall.  Most people, accidently or otherwise, give inaccurate histories usually underreporting. The results of this study appear to demonstrate this - the quantities here would describe intakes of:

28g cheese (a 2cm cube)/week,

1 slice bread with a very thin spread of fat/day,

1 portion of breakfast cereal/ week,

1/2 portion starchy carbs/day,

1 packet of crisps/week,

1 biscuit/day,

1 portion of meat or fish/day.

Other studies have produced more realistic data with diet diaries because these are acknowledged to be more accurate than FFQs e.g. Charnley M et al Pregnant Women living with obesity: a cross-sectional observational study of dietary quality and pregnancy outcomes. Nutrients 2011; Ainscough KM et al, An observational analysis of meal patterns in overweight and obese pregnancy. Irish Journal of Medical Science 2020.

Perhaps making the observation that intakes look inadequate along with a query about whether results might be more reliable should a different recording method be used in future studies is required for the discussion.

Minor comments:

4)      In the materials and methods, it would be useful to include a description of the 4 arms of the GLOWING study. Also, it would help to have more information on the intervention – the one-off full day of intensive face-to-face training for small groups of eligible midwives.

5)      Please spell out the acronym MET the first time it is used.

6)      Please state the time period (months/years) of the data collection.

7)      The paragraph in the discussion on the study being in the pre-COVID-19 introduces more speculation into a section which is already straining with under supported comments. I recommend only including this additional section if studies can be referenced which have quantified the impact of COVID-19 on diet, PA or weight during gestation of postnatal timepoints.    

Author Response

Heslehurst et al present a secondary analysis of a study of postnatal diet, physical activity (PA) and body weight in women with obesity from deprived areas in the UK. This manuscript extends that study (which was published in Nutrients) with follow-up data from the 3, 6, 9 and 12 months postnatal timepoints. These new data aimed to deepen the understanding of women’s diet, PA and weight changes in the postnatal period as it can be considered to also be a prenatal period for subsequent births. That time period is thus a window of opportunity for establishing improved health and lifestyle during the conceptional and gestation stages for subsequent children. Additionally, this type of study has not been performed before in the UK so the data will be valuable for that population.

  • Thank you for this positive feedback – we agree that this study addresses an important research gap in the UK and that the postnatal/preconception time period studied presents opportunities for maternal and infant health.

Major comments:

1)      As the authors repeatedly acknowledge, the study suffers from low subject numbers at every stage. While the study was undertaken with adherence to standard design guidelines, the low participant numbers make it difficult to justifiably extrapolate to larger populations. Statistical significance for novel observations is largely absent, even for the intriguing effect of gestational weight gain on postnatal weight loss (Table 3). Overall, this study supports the general view that future research should concentrate on obese women from deprived areas, but these results don’t add much to the current body of evidence.

  • As this was data from a pilot study and not powered to detect statistically significant results, and had a small sample size, we do not feel that statistical analysis to try and identify statistical significance in the results is methodologically appropriate.
  • We have described the study in the title as being pilot, and the aim refers to the exploratory nature, and we have discussed the limitations of small sample size and the lack of generalisability (i.e. ability to make inferences at the population level).
  • We disagree that this study does not add much to the current body of evidence. As discussed in the manuscript, there is an absence of UK based studies, and despite the small sample size and limitations of this study that are discussed, this paper provides important patterns in postnatal (and preconception) diet, physical activity and weight change data to inform future research needs for this highly deprived and high-risk population.  

2)      The conclusions state that the study has 'identified' that postnatal diet and PA are inadequate - I don't see that the evidence does more than 'suggest' this - even though we know it to be true from other research.

  • We have changed the wording to suggest rather than identified.
  • The context of this sentence is for “a highly deprived population of women living with obesity in the UK”, where we have discussed that there is a lack of other research.

3)      The reported dietary intake data are surprisingly low. Measuring dietary intake is notoriously difficult, one of the most important issues being the accuracy of dietary recall.  Most people, accidently or otherwise, give inaccurate histories usually underreporting. The results of this study appear to demonstrate this - the quantities here would describe intakes of:

28g cheese (a 2cm cube)/week,

1 slice bread with a very thin spread of fat/day,

1 portion of breakfast cereal/ week,

1/2 portion starchy carbs/day,

1 packet of crisps/week,

1 biscuit/day,

1 portion of meat or fish/day.

Other studies have produced more realistic data with diet diaries because these are acknowledged to be more accurate than FFQs e.g. Charnley M et al Pregnant Women living with obesity: a cross-sectional observational study of dietary quality and pregnancy outcomes. Nutrients 2011; Ainscough KM et al, An observational analysis of meal patterns in overweight and obese pregnancy. Irish Journal of Medical Science 2020.

Perhaps making the observation that intakes look inadequate along with a query about whether results might be more reliable should a different recording method be used in future studies is required for the discussion.

  • We agree that there are substantial challenges with collecting dietary intake data. We carried out extensive data collection for the wider pilot trial, and the participant burden for other methods of diet data collection would have been too high and likely resulted in even higher loss to follow up.
  • We used an FFQ that was developed for the EPIC study and adapted for a UK population. The FFQ has been validated against 24hr recalls and previously used to assess dietary intake in a large, multiethnic group of pregnant women living with obesity at multiple timepoints during pregnancy and in the postnatal period (Dalrymple KV et al., Nutrients. 2021).
  • The values reported are median and IQR for all diet items. We agree that the reported dietary intake is low for several items which suggests underreporting, which is common with self-report dietary assessment methods. The underreporting in the current study is in line with what has been published previously with the highest prevalence among those with pre-pregnancy overweight and obesity (e.g., McGowan CA, McAuliffe FM. Maternal nutrient intakes and levels of energy underreporting during early pregnancy. Eur J Clin Nutr. 2012 Aug;66(8):906-13. And McNitt KM, Hohman EE, Rivera DE, Guo P, Pauley AM, Gernand AD, Symons Downs D, Savage JS. Underreporting of Energy Intake Increases over Pregnancy: An Intensive Longitudinal Study of Women with Overweight and Obesity. Nutrients. 2022 Jun 1;14(11):2326).
  • We have added more critical discussion relating to this point to the discussion of strengths and limitations.

Minor comments:

4)      In the materials and methods, it would be useful to include a description of the 4 arms of the GLOWING study. Also, it would help to have more information on the intervention – the one-off full day of intensive face-to-face training for small groups of eligible midwives.

  • We have added some more brief information about the 2 arms (4 clusters) of the GLOWING study and intervention content – full details of the intervention are reported elsewhere and cited.

5)      Please spell out the acronym MET the first time it is used.

  • We have amended this.

6)      Please state the time period (months/years) of the data collection.

  • These dates are now added to the results

7)      The paragraph in the discussion on the study being in the pre-COVID-19 introduces more speculation into a section which is already straining with under supported comments. I recommend only including this additional section if studies can be referenced which have quantified the impact of COVID-19 on diet, PA or weight during gestation of postnatal timepoints.    

  • We have removed this discussion.